# Intracellular Trafficking and Distribution of Cd and InP Quantum Dots in HeLa and ML-1 Thyroid Cancer Cells

**DOI:** 10.3390/nano12091517

**Published:** 2022-04-29

**Authors:** Min Zhang, Daniel S. Kim, Rishi Patel, Qihua Wu, Kyoungtae Kim

**Affiliations:** 1Department of Biology, Missouri State University, 901 S National, Springfield, MO 65897, USA; zhang1996@live.missouristate.edu; 2Emory College of Arts and Science, Emory University, 201 Dowman Dr., Atlanta, GA 30322, USA; daniel.kim3@emory.edu; 3Jordan Valley Innovation Center, Missouri State University, 542 N Boonville Ave, Springfield, MO 65806, USA; rjpatel@missouristate.edu (R.P.); qwu@missouristate.edu (Q.W.)

**Keywords:** CdSe/ZnS, InP/ZnS, quantum dots, HeLa, ML-1, cancer, trafficking, distribution

## Abstract

The study of the interaction of engineered nanoparticles, including quantum dots (QDs), with cellular constituents and the kinetics of their localization and transport, has provided new insights into their biological consequences in cancers and for the development of effective cancer therapies. The present study aims to elucidate the toxicity and intracellular transport kinetics of CdSe/ZnS and InP/ZnS QDs in late-stage ML-1 thyroid cancer using well-tested HeLa as a control. Our XTT (2,3-bis-(2-methoxy-4-nitro-5-sulfophenyl)-2H-tetrazolium-5-carboxanilide) viability assay (Cell Proliferation Kit II) showed that ML-1 cells and non-cancerous mouse fibroblast cells exhibit no viability defect in response to these QDs, whereas HeLa cell viability decreases. These results suggest that HeLa cells are more sensitive to the QDs compared to ML-1 cells. To test the possibility that transporting rates of QDs are different between HeLa and ML-1 cells, we performed a QD subcellular localization assay by determining Pearson’s Coefficient values and found that HeLa cells showed faster QDs transporting towards the lysosome. Consistently, the ICP-OES test showed the uptake of CdSe/ZnS QDs in HeLa cells was significantly higher than in ML-1 cells. Together, we conclude that high levels of toxicity in HeLa are positively correlated with the traffic rate of QDs in the treated cells.

## 1. Introduction

Quantum dots (QDs), also known as heterogeneous, photostable semiconductor nanocrystals, are composed of a colloidal core surrounded by a shell coated with other surface modifications [1]. One or more shell layers are often used to encase the core of QDs to increase solubility in the aqueous medium, to lower intrinsic toxicity by reducing the leaching of heavy metals from the core, and to improve biocompatibility and surface chemo-physical properties for the attachment of QDs to other therapeutic and diagnostic molecules [2,3]. Unlike other engineered nanomaterials, the fluorescent emission and absorption spectra of QDs are size-dependent, and they are easily detected because they have high sensitivity, optical electro-chemiluminescence (ECL), and long-lasting photochemical properties, which make them widely used in cell labeling and imaging, immunohistochemistry, cancer-targeting therapies, and other biomedical applications [3,4,5,6]. QDs are resistant to photobleaching [7], which permits signals to be collected for longer periods of time, increasing image resolution [8]. Due to this property, QDs have been extensively used in biomedical imaging, including super-resolution imaging that permits imaging below the diffraction limit [9].

Extensive applications of QDs in cancer cells not only require clarification of their specific toxicity but also require a deep understanding of the interaction of QDs with the cellular system, including the uptake, and intracellular trafficking kinetics of QDs. It has been reported that the interaction between QDs and targeted cells involves different intracellular trafficking kinetics and pathways resulting in different toxicity profiles [10,11,12]. In fact, each type of QD exhibits a preferred endocytic pathway to enter cells depending on many factors, such as their surface coating and charge, serum, and cell type. Among these factors that could affect the endocytic pathways, the size of the QDs is the most important factor. It has been reported that clathrin-dependent endocytosis is the major endocytic pathway of QDs in mammalian cells and tissues [13,14]. Large particles with a size of 100 nm could be internalized via clathrin-dependent endocytosis, while particles with a size of 60–80 nm could be internalized via the caveolae-mediated endocytosis pathway [15].

The understanding of the biological interaction of the internalized QDs with cellular components such as macromolecules and organelles remains to be explored, and therefore, the underlying mechanisms for the internalized QD-based cytotoxicity are not fully understood. Among topics of QD-mediated impacts on cells, the toxicity derived from Cd- and InP-based QDs coated with ZnS layers has drawn much attention recently. According to the findings in previous studies on the toxic effects of CdSe/ZnS and InP/ZnS from Hens et al. and Davenport et al., both types of QDs elevate levels of apoptosis that leads to a viability reduction in HeLa cells [16,17]. However, the direct comparisons in toxicity resulting from the treatment of these two QDs in human cancer cell lines within one single study are scarce. As human cancer cells come in over 200 different varieties, it would be beneficial to investigate the toxic effects of CdSe/ZnS and InP/ZnS QDs using a wider spectrum of cells. Among those cancer cell lines, HeLa cells have been extensively studied by researchers around the world and explored for the study of the effects of QDs on mammalian cells [16,17,18,19,20,21,22]. Thyroid cancer has been predicted to become the fourth leading cause of new cancer diagnosis in the United States by 2030 [23]. ML-1 (ACC-464) cells are thyroid cancer cells, and they were derived from a 50-year-old patient who has dedifferentiated recurrent follicular thyroid carcinoma, which is radioactive iodine resistant. The recurrence of thyroid cancer, especially dedifferentiated thyroid cancer, has been a concern and attracted much attention, so the thyroid cancer cell line has often been used as the test model for biological and medical research [24]. Although ML-1 cells were considered a good model for toxicological investigations [24], the ML-1 cell model in the field of nanotechnology has been scarcely explored. So far, nothing is known about their interactions with QDs for medical practices. Therefore, we used both HeLa and ML-1 cell lines to see if these novel QDs have any differential toxic effects in those cell lines. We expect that the outcome of this research will provide more solid scientific evidence for safer and more efficient QDs-guided therapies for dedifferentiated thyroid cancers.

The first aim of the present study is to investigate the cytotoxicity of CdSe/ZnS and InP/ZnS QDs in ML-1 thyroid cancer cells. The second aim is to further elucidate the QD intracellular endocytic trafficking kinetics that may result in differences in sequential toxic effects. The present study fills a gap of knowledge in the field of nanomaterials, as we used four types of QDs and comprehensively and directly tested the toxic effects of Cd- and InP-QDs in two different cell lines, HeLa and ML-1, also revealing that ML-1 thyroid cancer cells are more resistant to these QDs when compared with HeLa cells. The rationale for the resistance is attributed to the fact that the transit rate of QDs in ML-1 cells was slower than in HeLa cells. The current study assessing the traffic rate of QDs in the endocytic organelles, including the endosome and lysosome, provides new insight into the molecular basis for the varying degree of severity affected by QDs in different types of cells. We also propose that the differences in uptake and trafficking rate of QDs in various cell lines are due to the metabolic rate of cells, and this underlying molecular mechanism will help guide further engineering of QD-based imaging probes and designing of QD-targeted cancer therapies for cancer diseases.

## 2. Materials and Methods

### 2.1. CdSe/ZnS and InP/ZnS Quantum Dots (QDs)

Water-soluble green Cadmium Selenide/Zinc Sulfide (green CdSe/ZnS: 6.1–9.5 nm) [16] green Indium Phosphide/Zinc Sulfide (green InP/ZnS: 3.7–5.2 nm) [17], red Cadmium Selenide/Zinc Sulfide (red CdSe/ZnS: 5–10 nm; Appendix A), and red Indium Phosphide/Zinc Sulfide (red InP/ZnS: 10–20 nm; Appendix A) were functionalized with carboxylic acid (NNCrystal; Fayetteville, AR, USA). The ZnS shell layer encases CdSe or InP core to achieve high optical stability, improve biocompatibility, and lower intrinsic toxicity. Green and red CdSe/ZnS and InP/ZnS were suspended in water (1 mg/mL). Red CdSe/ZnS and red InP/ZnS QDs were imaged using a JEOL 7900 (JEOL USA, Inc., Peabody, MA, USA) field emission scanning electron microscope (FESEM) equipped with a scanning tunneling electron microscopy (STEM) detector (Appendix A).

The hydrodynamic size of red CdSe/ZnS and red InP/ZnS QDs was characterized using dynamic light scattering (DLS). Red CdSe/ZnS and red InP/ZnS QDs were diluted to 0.1 mg/mL and were then tested by Colloid Metrix NANO-flex^®^ II (Colloid Metrix GmbH, Mebane, NC, USA) with a laser wavelength of 632 nm and scattering angle of 180° (Appendix A).

The peak positions and quantitative composition of red CdSe/ZnS (Appendix A) and red InP/ZnS (Appendix A) QDs were verified by X-ray photoelectron spectroscopy (XPS, Thermo Fisher Scientific Inc., Waltham, MA, USA). Approximately 0.5 mL of each of the QD stock solutions was pipetted onto a silicon substrate that was placed on a hot plate set to 50 °C. Each QD was pipetted at 50 µL increments where each aliquot was allowed to dry in between subsequent aliquots to form a film of QDs. The composition of each of the 2 QD materials (CdSe/ZnS and InP/ZnS) was evaluated by a Thermo Scientific Nexsa XPS (Thermo Fisher Scientific Inc., Waltham, MA, USA).

Also, the QD stock solutions were irradiated with a UV light source at 365 nm, and different degrees of red fluorescence were detected (Appendix A).

### 2.2. Cell Culture 

Two human cell lines, ML-1 thyroid cancer and HeLa-S3 cells, were used for this study. ML-1 thyroid cancer cells were derived from a patient suffering from dedifferentiated follicular thyroid carcinoma [25], and the HeLa-S3 cells were authenticated by Gentica Labs of Burlington, NC. To culture ML-1 or HeLa cells, cells were first taken from a −80 °C freezer and thawed in a warm water bath with a temperature of 37 °C until contents appear to contain hazy speckles. Then, thawed cells were mixed with 10 mL of pre-warmed Gibco Dulbecco’s Modified Eagle Medium (DMEM) media that was supplemented with 10% Fetal Bovine Serum (FBS) and 1% antibiotics (penicillin and streptomycin) in a 15 mL falcon tube. Cell mixtures were centrifuged at 400× *g* for 10 min to get the cell pellet. After centrifugation, the supernatant was removed from the 15 mL Falcon tube without disturbing the cell pellet. Thirteen milliliters of pre-warmed DMEM (with 10% FBS and 1% antibiotics) were added to the fully suspended cell pellet by pipetting. Last, the cell suspension was plated into a 75 cm^2^ culture flask, which was then stored in an incubator at 37 °C with 5% CO_2_/95% air atmospheric air. To maintain the cell growth in the culture flask, the media was changed every 3–5 days as needed. 

### 2.3. XTT-Proliferation/Viability Assay

The Cell Proliferation Kit II (XTT) viability experiment was separately conducted to test the cytotoxicity caused by CdSe/ZnS and InP/ZnS QDs in ML-1 thyroid cancer cells and HeLa cells, and the entire experiment for each cell line took three consecutive days to be completed. The protocol of XTT treatment on day 3 was based on the Biotium manufacturer’s protocol. On the first day, ML-1 or HeLa cells were re-cultured from a 75 cm^2^ culture flask and seeded into two flat-bottom 96-well plates at the density of 10,000 cells per well. Once cells were seeded, DMEM (with 10% FBS and 1% antibiotics) was added to each cell-seeded well to make a total volume of 100 µL per well. The 96-well plates with cells were then incubated in an incubator at 37 °C with 5% CO_2_/95% air atmosphere for 24 h. On the second day, the cell culture medium was removed and replaced with two different cell culture media: 100 µL DMEM. Then, 20 µL of CdSe/ZnS or InP/ZnS QDs were added into designated wells and serially diluted by a factor of 5 to give varying doses of QDs: 4.6 µg/mL, 28 µg/mL, and 167 µg/mL. Each concentration was repeated in triplicates. Twenty percent DMSO treated samples were used as a positive control in triplicate. The 96-well plates with treated cells were incubated at 37 °C with 5% CO_2_/95% air atmosphere for another 24 h. On the third day, XTT solution (XTT salt) and XTT Activation Reagent (PMS) were mixed in a ratio of 200:1 to make an XTT detection solution. Twenty microliters of the prepared XTT detection solution were added to each well, and the plate was incubated at 37 °C with 5% CO_2_/95% air atmosphere for 7 h. During this 7 h, the XTT agent turns into orange-colored formazan salts with the help of PMS oxidation. The XTT absorbance in each well was analyzed by a BioTek ELx880 Absorbance Microplate Reader (BioTek Instuments, Winooski, VT, USA) with Delta (A450–A630 nm) 7 h after the XTT treatment. Each concentration was repeated in triplicate. 

### 2.4. Reactive Oxygen Species (ROS) Assay

The ROS experiment was conducted over three days. On day one, 50,000 HeLa cells or ML-1 cells were seeded into 24-well plates. Then, the plates were incubated for 24 h. The next day, culture media in the plates were replaced with pre-warmed DMEM, and cells were treated with 28 µg/mL and 167 µg/mL of CdSe/ZnS or InP/ZnS QDs. After another 24 h incubation, the cells were washed twice with 1X PBS and detached from the well with 250 µL of trypsin with EDTA. All disassociated cells were transferred to Eppendorf tubes and centrifugated for 10 min at 400× *g*. The supernatant of each sample was removed after centrifugation, and the pellet was resuspended with dihydroethidium at the final concentration of 16 µM. All Eppendorf tubes were covered in foil and incubated in the dark at 37 °C for 30 min. The ROS level of each sample was measured by a flow cytometer (Attune NxT acoustic focusing cytometer, Life Technologies, Carlsbad, CA, USA) at 518 nm/606 nm (Excitation/Emission) for DHE.

### 2.5. Apoptosis Assay

The protocol of Apoptosis Assay was based on the manufacturer’s protocol (Thermo Fischer Scientific, Waltham, MA, USA, https://bit.ly/2mSZfhR, accessed on 16 August 2020). One day before the apoptosis experiment, 100 mL 10X Annexin V Binding Buffer was prepared by mixing 0.1 M HEPES, 1.5 M NaCl, 25 mM CaCl_2_, and 70 mL molecular grade sterile water thoroughly together. On the first day of the experiment, 50,000 HeLa cells or ML-cells were seeded into 24-well plates. Cell culture media in each well was replaced with DMEM after 24 h incubation, and cells were treated with 167 µg/mL of green and red CdSe/ZnS or InP/ZnS QDs. The plate was incubated for another 24 h. On day 3, to create 20 mL of 1X Annexin V Binding Buffer, 2 mL 10X Annexin V Binding Buffer and 18 mL 1X PBS were mixed. All wells were washed with 1X PBS twice, and 250 µL non-EDTA Trypsin was added to each well to detach cells. After 20-min incubation, 250 µL of fresh DMEM was added to each well to neutralize the non-EDTA Trypsin. All disassociated cells were transferred to Eppendorf tubes and centrifuged for 10 min at 1000× *g*. The supernatant was removed from each tube after centrifugation. The pellet of untreated control was resuspended with 500 µL 1X PBS, and the pellets of 3 blanks were resuspended with 100 µL 1X Annexin V Binding Buffer, 5 µL Annexin V-APC, and 5 µL propidium iodide (PI), respectively. All treated samples were resuspended with 100 µL 1X Annexin V Binding Buffer, 5 µL Annexin V-APC, and 5 µL PI. After 30-min incubation, another 400 µL of 1X Annexin-V binding buffer was added to the blank that contains Annexin V-APC and all treated samples. The apoptosis level of each sample was measured by the Attune NxT acoustic flow cytometer (Life Technologies-Thermo Fisher Scientific, Waltham, MA, USA) based on excitation/emission filter sets of 650/660 nm and 533/616 nm for Annexin-V-APC and Propidium Iodide, respectively.

### 2.6. QD Colocalization Analysis

The day before the experiment, HeLa and ML-1 cells were seeded (HeLa: 10,000 cells/well; ML-1: 5500 cells/well) into 18-well plates in a medium to optimize each well with 30–40% of confluence. The next day, cell culture media in each well was replaced with DMEM (serum+) and DMEM (serum−). Cells were treated with 14 µL (1.4 × 10^6^ particles/mL) Rab5a-TagRFP, 14 µL (1.4 × 10^6^ particles/mL) Rab7a-TagRFP and 20 µL (2 × 10^6^ particles/mL) LysoView 540 (Thermo Fisher Scientific, Waltham, MA, USA) in triplicate, and they were incubated at 37 °C with 5% CO_2_/95% air atmosphere for 3 h. Then, 5 µL of freshly homogenized green CdSe/ZnS QDs were applied to treated cells, and cells in the first 18-well plate were incubated at 37 °C with 5% CO_2_/95% air atmosphere for 24 h. Cells in another 18-well plate were incubated at 37 °C with 5% CO_2_/95% air atmosphere for 48 h. On the third day, the culture media of the first plate was removed and cells were washed three times with 1X PBS. After washing, cells in PBS were visualized immediately under confocal fluorescent microscopy. On the fourth day, the culture media of the second plate was removed and cells were washed with 1X PBS three times. Cells were visualized immediately for 48 h incubation of QDs under confocal fluorescent microscopy.

### 2.7. Quantification of CoLocalization with Endosomal Rabs and LysoView

The extent of colocalization of QDs to endosomal Rabs and LysoView 540 was expressed by the mean of Pearson’s correlation coefficients using the ImageJ plugin JACoP. Briefly, each cell labeled with green QDs and a red internal organelle reference marker (Rab5a-TagRFP, Rab7a-TagRFP, or LysoView 540) was cropped to a size of 250 by 250 pixels for each treatment group in triplicate. Two images with green QDs or a red reference marker for a single cell were opened at one time, and then they were converted to RGB format. The plugin JACoP panel was opened. Image A and B were automatically selected once opening the two images with green QDs or a red reference marker for a single cell. Pearson’s Coefficient and Coste’s Automatic Threshold were selected on the JACoP panel and Analysis was clicked to analyze the colocalization area on the images. Once the colocalization analysis was done, a Log Panel with Pearson’s Coefficient r-value, which quantifies the degree of colocalization between green QDs and a red internal organelle reference marker, was provided. 0 = no colocalization; 1 = full colocalization. 

### 2.8. Confocal Fluorescent Microscopy

An Olympus IX-81 (Olympus IE, Waltham, MA, USA) inverted spinning confocal microscope equipped with an ImagEM camera was used to take images of intracellular endocytic kinetics. For colocalization imaging of both QDs, green (488 nm) and red lasers (561 nm) were turned on. C9100–13 S/N: 180663 camera and GFP. SDC of SDC single filter was selected on the Focus Window panel. The 10× lens of a confocal microscope with a lamp of approximately 25% was used to focus on mammalian cells in the 18-well slide plate, and the 40× lens with a lamp of approximately 32% was then used to focus on the organelles inside of mammalian cells. Once cells were focused under a confocal microscope, the Alt and Live on the Focus Window panel were closed. The Capture Window was opened, and the QDs tracking images were captured using SDC single filter with 200 ms GFP.SDC and RFP.SDC and 200 intensities through ImageEM camera. 

### 2.9. QD Internalization for ICP-OES

Inductively coupled plasma optical emission spectroscopy (ICP-OES, Thermo Scientific iCAP7400 Duo, Waltham, MA, USA) was used to determine the amount of internalized green CdSe/ZnS QDs in ML-1 and HeLa cells 48 h after the treatment. ML-1 or HeLa cells were seeded in two 6-well plates at a density of 500,000 cells/well. After cell seeding, the 6-well plates with cells were stored in the incubator at 37 °C with 5% CO_2_/95% air atmosphere for 24 h. On the second day, cell culture media was removed and replaced with DMEM (serum+) and DMEM (serum−). Fifty and one hundred microliters of homogenized green CdSe/ZnS QDs were added into the designated wells to give 50 µg/mL, 50 µg/mL, and 100 µg/mL concentrations, respectively. The cells in untreated wells were used as non-treated control. Each treatment was repeated in triplicate. The plates were incubated at 37 °C with 5% CO_2_/95% air atmosphere for 48 h. On the fourth day, the culture medium was removed, and cells were washed three times with 1X PBS. The cells in each well were detached by 500 µL trypsin with EDTA, which then was neutralized by 1 mL DMEM with 10% FBS. After neutralization, cell mixtures were transferred into the Eppendorf tubes and centrifugated at 1000× *g* or 10 min to get the pellet. The supernatant of each sample was removed after centrifugation. The pellet was washed twice by centrifugation in 1X PBS. The number of cells was counted under microscopy by using a hemocytometer, and each cell sample ultimately contained 317,500 cells/mL for ML-1 and 92,500 cells/mL for HeLa. Samples were wet-digested by adding 1 mL of concentrated HCl-HNO_3_ 3:1 (*v*/*v*) overnight, and they were then diluted to 10 mL with DI water prior to the test.

A total of 19 elements (including Cd and Zn) were analyzed by ICP-OES (Thermo Scientific iCAP7400 Duo, Waltham, MA, USA) according to EPA method 200.7 [26] with minor modifications. Reagent blanks and standard solutions with known amounts of standard mixtures were used for quality control (QC) purposes. The method was validated with respect to method detection limit (MDL), linearity range, and repeatability. The accuracy of the test method was monitored by analyzing QC standards for every 10 samples.

### 2.10. QDs Release for ICP-OES

The amount of green CdSe/ZnS QDs excreted by ML-1 and HeLa cells was quantified by ICP-OES. ML-1 or HeLa cells were seeded in two 24-well plates at a density of 50,000 cells/well. The plates were incubated at 37 °C with 5% CO_2_/95% air atmosphere for 24 h. On the next day, cell culture media was removed and replaced with DMEM (serum+) and DMEM (serum−). Cells in designated wells were treated with 50 µL and 100 µL sonicated green CdSe/ZnS QDs. Untreated cells were used as non-treated control. To this end, cells were labeled with green CdSe/ZnS QDs at 50 µg/mL and 100 µg/mL concentrations for 24 h. After treatment, the 24-well plates with cells were stored in the incubator at 37 °C with 5% CO_2_/95% air atmosphere for 24 h for QDs internalization. On day 3, the cell medium was collected and then diluted to 3 mL with DI water. This resulting solution was used to determine the amount of internalized green CdSe/ZnS QDs in cells. Then, cells were washed twice with 1X PBS and supplemented with fresh DMEM (serum+) and DMEM (serum−). Culture supernatant samples were collected into 15 mL conical centrifuge tubes at 24 h and then diluted to 3 mL with DI water. Samples were wet-digested by adding 1 mL of concentrated HCl-HNO_3_ 3:1 (*v*/*v*) overnight, and they were then diluted to 10 mL with DI water prior to the test. Samples were analyzed by ICP-OES using the same method as described previously (Section 2.9). 

### 2.11. Statistical Analysis

GraphPad Prism 8.0 (GraphPad Software, Inc., San Diego, CA, USA) was used to statistically analyze the data of all experiments. One-Way ANOVA and Dunnett’s multiple comparisons were used to assess variance between control and treatment groups. On all prism graphs, each sample is represented by an average of three replicates and has error bars representing standard deviation. Statistically significant data is represented on graphs as * *p* < 0.05, ** *p* < 0.01, *** *p* < 0.001, **** *p* < 0.0001. 

## 3. Results

### 3.1. Effect on Cell Viability When Treated with CdSe/ZnS and InP/ZnS QDs

It has been reported that the cell viability of HeLa was decreased significantly by 69 µg/mL of InP/ZnS and CdSe/ZnS QDs [16,17]. There are abundant Cd and InP QD-related studies using the HeLa human cervical cancer cell model to test the toxicity effects of QDs, while no QD-related studies have been conducted with the ML-1 thyroid cancer cell model. As such, these cells were treated for 24 h with these QDs with varying concentrations from 4.6 to 167 µg/mL via a serial dilution by a factor of 5. After 7 h of incubation with XTT activation reagent, ML-1 thyroid cell viability was determined. It was found that there was no QD-induced viability defect in ML-1 cells (Figure 1A,B). In contrast, HeLa cell viability, which is correlated with measured absorbance levels, was reduced by approximately 30% when treated with 167 µg/mL of green CdSe/ZnS QDs, compared with the non-treated control (Figure 1C). It appears that red Cd QD did not affect HeLa cell growth (Figure 1C). Green InP/ZnS QDs reduced HeLa cell viability by approximately 40% at its 167 µg/mL concentration, while 167 µg/mL of red InP/ZnS QD caused approximately 30% reduction in cell viability when compared to that of the non-treated control (Figure 1D). Positive control experiments were performed using DMSO (5–20%) (Figure 1). 

To assess the effect of green and red CdSe/ZnS and InP/ZnS on non-cancerous cells, mouse-derived fibroblast cells were subjected to the same viability assay. Interestingly, none of these QDs affect cell viability based on minor differences in the average values between treated groups and the non-treated control. Nevertheless, the statistical analysis indicated that Cd QDs, regardless of their emission status, was slightly more toxic to mouse fibroblast cells than InP QDs (Figure 1E,F). Overall, the results indicate that HeLa cells were more sensitive to Cd and InP QDs compared with ML-1 cells because CdSe/ZnS and InP/ZnS induced statistically significant cell viability reduction in HeLa cells. 

### 3.2. The Viability Reduction of HeLa Cells Was Not Due to Oxidative Stress

Elevation of ROS is recognized as the toxic by-product of cellular metabolism, and it plays a key role in disturbing the survival of the cell in stressful environments [27]. It was reported previously that the decreased viability of HeLa cells treated with InP/ZnS and CdSe/ZnS was induced by a high level of superoxide production [28]. Excessive production of ROS can cause different degrees of oxidative damage and ultimately cellular death [28], and therefore, we reasoned the possibility that the reduced cell viability (Figure 1) might be caused by an increase in oxidative stress in HeLa cells treated with CdSe/ZnS and InP/ZnS QDs. We then measured levels of ROS produced in cells treated with concentrations of QDs ranging from 28 µg/mL to 167 µg/mL CdSe/ZnS and InP/ZnS. Only 10% DMSO was used in this ROS measurement because it significantly reduced cell viability as effectively as 20% DMSO (Figure 1). Samples were dyed with dihydroethidium (DHE) to measure the production of superoxide radicals. Interestingly, there was no statistical difference in superoxide production between QD-treated ML-1 thyroid cancer cells and non-treated control (Figure 2A,B). Red Cd and InP QDs and green Cd and InP QDS with a concentration of 167 µg/mL did not cause a significant increase in ROS production in HeLa cells (Figure 2C,D). As a result, the viability reduction of HeLa cells was not from the oxidative stress caused by QDs. The superoxide level of ML-1 cells did not show statistically significant differences compared to NTC groups, which is consistent with XTT cell viability assay results (Figure 1).

### 3.3. The Viability Reduction of HeLa Cells Was Due to Early and Late Apoptosis

The HeLa cell viability reduction observed via the cell viability assay (Figure 1) was not due to superoxide (Figure 2), and then we tested the possibility that the reduction of HeLa cell viability might be due to an elevation of apoptosis. We tested for induction of apoptosis in cells treated with 167 µg/mL of CdSe/ZnS and InP/ZnS QDs. For this, Annexin-V-APC and propidium iodide-stained cells were used to measure the extent of early and late apoptosis induced, respectively. In ML-1 cells, there was no increase in early apoptosis, but 167 µg/mL of red CdSe/ZnS and green InP/ZnS increased, albeit marginally, levels of late apoptosis, compared with the non-treated control (Figure 3A,B). In HeLa cells, all QDs caused elevation of early and late apoptosis (Figure 3C,D). Of these, green InP/ZnS QDs were the least effective in inducing late apoptosis (Figure 3D). These results indicate that the statistically significant cell viability reduction (Figure 1) could mainly be attributed to elevated late apoptosis in HeLa cells, not superoxide stress (Figure 2).

### 3.4. QD Transportation Rate in HeLa Is Faster Than That in ML-1

Based on our findings that the same quantum dots (green CdSe/ZnS) have different degrees of impact on different cells (Figure 1, Figure 2 and Figure 3), with especially ML-1 cells being more resistant to these QDs, we then hypothesized that the transition speed of QDs from the early endosome to the late endosome and then to the lysosome in HeLa cells will be faster than compared with ML-1 cells. We decided to use green CdSe/ZnS QDs from the QDs pools for future experiments since they are potent QDs in reducing cell viability (Figure 1, Figure 2 and Figure 3) and increasing cell apoptosis (Figure 3). We transduced HeLa cells with baculovirus harboring Rab5a-TagRFP recombinant DNA and Rab7a-TagRFP recombinant DNA to label early and late endosomes, respectively [29]. In addition, LysoView 540 dye was used to stain the lysosome [30]. 

HeLa cells expressing Rab5a-TagRFP were treated with 5 µg/mL green CdSe/ZnS QDs for 24 h or 48 h before imaging using a spinning disk confocal microscope. At 24 h, QDs in cells grown with serum showed a higher level of colocalization with Rab5a-TagRFP labeled early endosomes (Figure 4A—top). An exemplary spot that does not show colocalization between QDs and an early endosome was indicated as an arrow (Figure 4A—bottom). It appears that QDs showed high levels of colocalization with Rab7a-TagRFP labeled late endosome after 24 h QD treatment, regardless of whether serum is present or not in the medium (Figure 4B). Furthermore, the colocalization of QDs with LysoView 540 labeled lysosome were higher in both cell groups grown in serum positive and serum negative conditions (Figure 4). Together, these results suggest that QDs reach the late endosome and the lysosome within 24 h and that QDs transit toward these organelles in cells grown in the medium lacking serum is slightly faster than in cells cultured with serum.

After 48 h of treatment with QDs, the extent of colocalization between QDs and the early endosome, in both serum lacking and serum present conditions, decreased (Figure 5A), with PCCVs of 0.6860 and 0.5503, respectively, which are lower than the corresponding values measured at 24 h. In contrast, levels of colocalization between QDs and late endosomes increased slightly over the corresponding levels at 24 h (Figure 5B). The QD colocalization with the lysosome at 48 h was as consistently high as the level of colocalization at 24 h (Figure 5C). Therefore, we concluded that more QDs travel to the late endosome and the lysosome at 48 h when compared with cells treated with QDs for 24 h.

One interesting fact observed while organizing the colocalization data was that cells cultured in the medium lacking serum displayed higher QD fluorescence intensities over cells grown with serum (Figure 4C and Figure 5C).

We then carried out the same subcellular colocalization experiments with ML-1 cells (Figure 6 and Figure 7). According to the Pearson’s Coefficient Test, QDs in HeLa cells grown with serum showed a higher mean Pearson correlation coefficient value (PCCV) of 0.8025 than that of QDs in cells grown without serum (PCCV = 0.5810) at 24 h of treatment. In both cell conditions, the PCCV value of QDs colocalization in lysosomes was higher than 0.9. At 48 h, PCCV values of QDs colocalization in early endosomes in both cell conditions were 0.6860 (serum−) and 0.5503 (serum+) (Figure 8). Unlike HeLa cells, in which QDs displayed more colocalization with the late endosome and the lysosome (Figure 4, Figure 5 and Figure 8) at both 24 and 48 h, QDs in ML-1 cells both at 24 and 48 h showed high levels of colocalization with all three organelles, the early endosome, the late endosome, and the lysosome, with Pearson’s Coefficient values higher than 0.9 (Figure 9). Therefore, regardless of serum in the culture media, the overall transit rate of ML-1 cells was slower than HeLa cells.

### 3.5. HeLa Cells Accumulate More Internalized QDs

We quantified the amount of internalized green CdSe/ZnS QDs by ML-1 and HeLa cells treated with 50 µg/mL of green CdSe/ZnS QDs with or without serum as well as cells with 100 µg/mL cultured in serum− media for 48 h using ICP-OES to compare the uptake rate of QDs in ML-1 and HeLa cells. After detaching cells, the total number of cells from each treatment was counted using a hemocytometer (Figure 10A,B). The results showed that the mean of viable ML-1 cells treated with green CdSe/ZnS QDs in either serum+ or serum− medium was not statistically significantly different from that of NTC groups (Figure 10A), which is consistent with our XTT results (Figure 1). However, the mean of viable HeLa cells with 50 µg/mL of QDs in serum− was 53.4% lower than that of cells treated with 50 µg/mL of QDs in serum+, which indicates that HeLa cells in serum− medium were less viable in response to green CdSe/ZnS QDs. With 100 µg/mL of CdSe/ZnS in serum−, the mean of viable cells was 37% lower than that of 50 µg QDs treated HeLa in serum− (Figure 10B). Then, these treated cells were analyzed by the ICP-OES test (Figure 10C,D). For ML-1 cells treated with 50 µg of QDs in serum−, the concentration of internalized Cd was 38% higher than that of ML-1 cells treated with 50 µg/mL QDs in serum+. ML-1 with 100 µg/mL QDs in serum− had 52% higher internalized green CdSe/ZnS QDs than ML-1 with 50 µg/mL QDs in serum−. NTC showed 0 ppm (Figure 10C). For HeLa, cells treated with 100 µg/mL QDs in serum− had 29% higher internalized green CdSe/ZnS QDs compared to that of cells treated with 50 µg/mL QDs in serum−. Importantly, HeLa cells treated with 100 µg/mL QDs in serum− internalized 6 times more CdSe/ZnS QDs than that of ML-1 treated with 100 µg/mL QDs in serum− (Figure 10D). All these results indicate that HeLa cells had a higher uptake rate of QDs than ML-1 cells, especially in serum− medium. 

### 3.6. HeLa in Serum + Medium Secreted More CdSe/ZnS QDs That Were Taken up Previously

We also quantified the amount of secreted CdSe/ZnS QDs by ML-1 and HeLa cells that were treated with 50 µg/mL of green CdSe/ZnS QDs with or without serum as well as cells with 100 µg/mL cultured in serum− media for 24 h using ICP-OES test. The cell culture media were collected (Figure 11) and subjected to ICP-OES to quantitate the amount of cadmium in the media, which provides an insight into the internalization dynamics of QDs under different growing conditions. The results showed that HeLa cells that were treated with 50 µg of green CdSe/ZnS QDs in serum− media took up more QDs than HeLa cells with the same amount of QDs in serum+ media (Figure 12A, Left, comparing the second bar and third bar), which is consistent with our fluorescence microscopy assay results. After removing the media containing QDs, we applied fresh, QD-free media to the culture and collected the media at different time points to measure the amount of cadmium secreted or released by the cell culture (Figure 11). Interestingly, HeLa cells treated with 50 µg green CdSe/ZnS QDs in serum+ media released more QDs than cells treated with 50 µg green CdSe/ZnS QDs in serum− media after 24 h of incubation with the fresh media (Figure 12B, right, comparing the second and third bar). Together, QDs were taken up more readily and secreted less as cells were placed under a starved condition.

## 4. Discussion

QDs are being explored for bioimaging and therapeutic purposes to a greater depth due to their superior photoemission and photostability characteristics [10,31]. For instance, recent literature has demonstrated that in vitro biomolecular profiling of cancer biomarkers, in vivo tumor imaging, and dual-functionality tumor-targeted imaging and drug delivery have been developed by using the superior photophysical characteristics of QDs [32]. However, the intracellular endocytic kinetics and subsequent cytotoxicity of QDs are poorly understood, which has raised concerns over their impact on human health [31,33]. Among human cancer cell lines, HeLa cells have been extensively explored for the study of the effects of QDs on mammalian cells. Additionally, in 2009, Pilli et al. found that ML-1 thyroid cancer cell, which is a type of progressive radioactive iodine-resistant thyroid cancer, is a great model for toxicology investigation [24]. To date, the existing therapies for ML-1 cells are not very effective, and little is known about the effects of QDs on ML-1 cells. To this end, the present study aims to fill the gap in this knowledge by investigating the intracellular endocytic kinetics of QDs and their subsequent toxicity profiles in ML-1 cells, by using well-tested HeLa cells as a control. This study revealed that green CdSe/ZnS QDs have different endocytic kinetics in different human cell lines, which accordingly appears to be accountable for the severity of their toxicity levels. These findings provide new insights into the cell-specific dynamics of QD endocytosis that could be exploited to improve the designing, efficacy, and specificity of QD-related drug delivery and targeting therapies.

Moreover, the present study provides valuable impacts on the field of QD-mediated nanotoxicology. Compared to other studies, we comprehensively tested the toxicity of Cd- and InP-QDs in cells, including green and red CdSe/ZnS and green and red InP/ZnS QDs. InP-QDs have long been recognized as a safer alternative to Cd-QD [34,35], and the toxicity of QDs was proposed to be size-dependent. Inconsistent with these, the IC50 values and ranking of toxicity of QDs (Table 1) in our study demonstrated that InP/ZnS QDs were more toxic than CdSe/ZnS QDs in HeLa cells. In addition, red InP/ZnS QD (10–20 nm), which is the largest QD in this study, caused more toxic effects than smaller green CdSe/ZnS (6.1–9.5 nm) and red CdSe/ZnS QDs (5–10 nm). These findings suggest that InP-QDs are not a safer alternative to Cd-QDs, at least in the cervical cancer cells, thus providing a new idea for the design of cancer therapies with InP/ZnS QDs. Second, we propose here that the metabolic rate of a cell type is directly correlated with the internalization and trafficking rate of QDs. Therefore, the metabolic rate/QD trafficking rate of a cell type should be an important parameter prior to designing an effective concentration of QD treatment and applications to fluorescence intracellular imaging, labeling, and biosensors. Furthermore, several studies have focused on studying the trafficking and distribution of QDs in endosomes and lysosomes in cells at certain time points. In contrast, our study compared the transit rate of QDs between endosomes and lysosomes, particularly in different cell lines. We also found that the trafficking rate of QDs towards the lysosome in HeLa cells is faster than that of ML-1 cells, most likely due to a fast metabolism taking place in HeLa cells, which is consistent with our second unique contribution mentioned above.

### 4.1. Faster Transporting Rate of QDs in HeLa Cells

To trace the transportation of QDs in both ML-1 and HeLa cells, they were treated with green CdSe/ZnS QDs for 24 and 48 h to quantify the subcellular localization of QDs. Four hypothetical models (Figure 13, Figure 14, Figure 15 and Figure 16) illustrate QDs’ intracellular localization and transportation among organelles in HeLa and ML-1 cells, 24 h and 48 h after the treatment of green CdSe/ZnS QDs, respectively. 

After 24 h of the treatment, QDs already colocalized with early and late endosomes and lysosomes in both ML-1 and HeLa cells grown in the medium with or without serum (Figure 8 and Figure 9). According to the colocalization assay in HeLa cells (Figure 8), higher levels of QD colocalization with early endosomes and late endosomes were detected in cells with serum than in cells without serum after 24 h of green CdSe/ZnS QD treatment, indicating that the transit rate of the endocytosed QDs from the early endosome towards the late endosome (labeled as T2, Figure 13) in HeLa cells with serum is slower than their counterpart rates in HeLa cells without serum. Additionally, it appears that the transit rate of the QDs from the late endosome towards the lysosome (labeled as T3, Figure 13) in HeLa cells with serum is slower than their counterpart rates in HeLa cells without serum due to a slightly lower level of QD colocalization with the late endosome in the serum-lacking condition. 

In light of the finding that the higher amounts of green CdSe/ZnS QDs were internalized into HeLa cells without serum than HeLa cells with serum after 24 h of treatment, and therefore, the uptake and internalization of QDs by HeLa cells grown without serum are more efficient compared to that of HeLa cells with serum. Together, we surmise that the transit rate of endocytosed QDs from the endocytic vesicles towards the early endosome (labeled as T1, Figure 13) in HeLa cells with serum is slower than their counterpart rate in HeLa cells without serum. Overall, we propose a model in which HeLa cells without serum have faster QD delivery rates (T1, T2, and T3) than those in cells with serum (Figure 13).

At 48 h, more QDs were found to be localized to the early endosome in HeLa cells with serum than in HeLa cells without serum (Figure 8), as found at 24 h. As such, the transit rate towards the late endosome (T2) in the HeLa cells with serum is slower than the counterpart rate in the HeLa cells without serum (Figure 15). The levels of colocalized QDs with both the late endosome and the lysosome in HeLa cells with serum were similar to those of HeLa cells without serum (Figure 8), indicating that the transit rate of QDs towards the lysosome (T3) in HeLa cells with serum is similar to that in HeLa cells without serum. 

The ICP-OES results displayed that HeLa cells without serum take up more CdSe/ZnS QDs than HeLa cells with serum 48 h after the treatment. This result indicates that the uptake and internalization of QDs in HeLa cells without serum is faster than that of HeLa cells with serum. Based on this, we conclude that the transit rate of QDs towards early endosome (T1) in HeLa cells without serum is faster compared to the counterpart rate in HeLa cells with serum at 48 h of the treatment. Therefore, even though the transit rate of T3 in HeLa cells without serum is similar to the transit rate of T3 in the HeLa cells with serum, HeLa cells grown without serum still have overall faster transit rates compared to the HeLa cells with serum 48 h after the treatment because HeLa cells with serum have faster transit rate in T1 and T2.

For ML-1 cells, the levels of QD colocalization in early and late endosomes and lysosomes in cells grown with serum were similar to those of cells grown without serum at 24 h of the treatment (Figure 9), pointing that the transit rates in T1, T2, and T3 in ML-1 cells with serum were similar to their counterpart rates in ML-1 cells without serum (Figure 14).

The ICP-OES results at 48 h of the treatment showed that more QDs were internalized into ML-1 cells grown without serum compared to ML-1 cells grown with serum. One can conclude that the uptake and internalization of green CdSe/ZnS QDs were more efficient in ML-1 cells without serum than in ML-1 cells grown with serum, which is consistent with the results obtained with HeLa cells. Hence, it was predicted that the transit rate of QDs in T1 is faster in ML-1 cells grown in the media lacking serum than in ML-1 cells grown with serum. Indeed, it was found that lower levels of QD colocalization were detected in the late endosome in ML-1 cells in the media without serum compared to those were detected in the late endosome in ML-1 cells grown with serum, while levels of QD colocalization in the early endosome and lysosome were similar at 48 h. Therefore, we conclude that the transit rate of QDs (T2) in ML-1 cells grown with serum is similar to that of ML-1 cells grown in the media without serum at 48 h. Additionally, the transit rate of QDs in T3 in ML-1 cells grown with serum is slower than that of ML-1 cells grown in the media lacking serum (Figure 16). In summary, the QD transit model in ML-1 cells proposes that the overall transit rate of QDs in ML-1 cells grown in the media without serum is faster compared to that of ML-1 cells grown with serum.

It is worth noting that, unlike HeLa cells, the colocalization of QDs with early and late endosomes and lysosomes in ML-1 cells at both 24 and 48 h of the treatment displayed Pearson’s Coefficient values higher than 0.9. This result indicates that the transit rate of QDs in ML-1 cells was slower than that in HeLa cells. This conclusion is further backed up by the results in which HeLa cells took up and contained 6 times more CdSe/ZnS QDs than ML-1 cells after 48 h of QD incubation (Figure 10). The more QDs in the cells, the more harmful effects they can cause. Based on this notion, the observation of lower cell viability and higher apoptosis levels found in HeLa than in ML-1 cells is attributed to a higher level of QD internalization and the subsequent damage by internalized QDs. Overall, the transit models (Figure 13, Figure 14, Figure 15 and Figure 16) illustrate that when cells are under starvation conditions (media without serum), the QD transit rate is faster than that of cells under normal conditions (media with serum). 

### 4.2. Comparison of QDs Traffic Dynamics

Cell-Type Effects: The QD transit models (Figure 13, Figure 14, Figure 15 and Figure 16) propose faster transit rates of QDs in HeLa cells compared to that of ML-1 cells, suggesting the transit rate of QDs is varied in different cell lines. This observation is in good agreement with another study, in which InP/ZnS QDs were more easily endocytosed by human lung cancer cells (HCC-15) than by Alveolar type II epithelial cells (RLE-6TN) because HCC-15 cells showed higher QD uptake or endocytic efficiency [36]. Interestingly, a study conducted by Peuschel et al. found that the intracellular localization and translocation even varied according to the differentiation status of cells. When the differentiated Caco-2 human intestinal cells were exposed to CdSe/ZnS QDs, no QDs were detected in the cell lumen using ICP-OES. Cytotoxicity of QDs was not detected, either. In contrast, CdSe/ZnS QDs could be detected in the cell lumen of undifferentiated Caco-2 cells, and they were localized in endosomes or lysosomes [37]. Without controversy, the present study showed a higher amount of green CdSe/ZnS QDs were found in HeLa cells than found in ML-1 cells. Interestingly, a study conducted by Liao et al. found ZnO nanoparticles can reduce cell viability and elevate ROS in cancer cells more than in normal cells due to the faster metabolic rate of cancer cells [38]. Based on the literature, the doubling time of the HeLa cell (approximately 10 h) [39] is much faster than that of the ML-1 cell (approximately 4–7 days) [25]. Taken together, these findings hint that green CdSe/ZnS QDs used in the present study were internalized with faster transit rates in HeLa cells due to a faster metabolic rate in HeLa cells than ML-1 cells. 

Cells grown in media lacking serum take up more QDs: The present study found that the intracellular endocytic transportation of QDs was also affected by the surrounding environment of QDs, such as serum in the culture media. Likewise, Damalakiene et al. observed the distribution of carboxylated red CdSe/ZnS QDs coated with PEG (Polyethylene Glycol) layer inside NIH3T3 cells. After 24 h of QDs incubation, QDs and the lysosome were fully colocalized with each other in media with serum [40]. In line with the present study, carboxylated green CdSe/ZnS QDs were colocalized with the lysosome both in HeLa and ML-1 cells in the media with serum at 24 h of QD treatment. However, they found that QDs were not colocalized with the lysosome in a medium without serum 24 h after the treatment [40], which is not consistent with the present study. In contrast, the present study demonstrated that carboxylated green CdSe/ZnS QDs already colocalized with lysosomes, regardless of whether serum is present or not (Figure 8 and Figure 9). Furthermore, cells in the media without serum displayed an overall higher efficiency of QD uptake and internalization compared to the cells grown with serum at 24 h. A similar result was reported by a study conducted by Lesniak et al. They revealed that silica QDs in the medium without serum have stronger adhesions to the cell membrane than in the medium with serum, leading to a higher internalization efficiency of the QDs in the medium without serum [41]. Consistently, another study also reported that the cellular uptake of CdSe/ZnS QDs was inhibited by media with serum [42]. Therefore, the consensus theme in QD internalization is that cells without serum take up QDs more effectively, but it is thought that the opposite result by Damalakiene et al. might be due to the use of QDs with a unique coating material including PEG. 

The Effects of Dose of Treatment: It has been reported that the intracellular trafficking route of QDs is dose-dependent. Carboxylic-coated CdSe/ZnS QDs at a concentration of 2 nM were colocalized with the early endosome in HEK cells, and then the level of colocalization gradually decreased over time. When the concentration was increased to 20 nM, although QDs were aggregated, many unaggregated QDs were found around the periphery of the nuclear membrane [31]. Damalakiene et al. found that carboxylated CdSe/ZnS QDs at the concentration of 10 nM were either adherent to the membrane or concentrated in a perinuclear region in fibroblasts [40]. Another study found that 4 nM carboxylated CdSe/ZnS QDs were aggregated on the surface of the fish embryo [43]. Xiao et al. revealed that a lower concentration (0.8 nM) of carboxylated CdSe/ZnS QDs were internalized and found in the lysosome [33]. Similarly, the current study revealed that 0.1 nM (5 µg/mL) of carboxylated green CdSe/ZnS QDs were colocalized with endosomes and lysosomes in both ML-1 and HeLa cells 24 h after treatment (Figure 8 and Figure 9). Based on all these observations, cells are capable of directing a lower concentration of QDs, down to 0.01 nM, towards endosomes, while with a higher concentration of QDs, up to 20 nM, cells can accumulate QDs around the perinuclear region. The present study provided a valuable insight into studying the intracellular destination of QDs in cells using CdSe/ZnS QDs with the lowest concentration (0.1 nM). 

The Effects of Time of Treatment: It was found that the extent of colocalization of carboxylated green CdSe/ZnS QDs with early endosomes in HeLa cells decreased over time, while the level with late endosomes slightly increased over time in the present study (Figure 8 and Figure 9). Importantly, in agreement with the present study, another study found that the internalized carboxylated CdSe/ZnS QDs colocalized with the early endosome at 1 h of the treatment of QDs. When the culture time increased, the colocalization with the early endosome gradually decreased, and the QD colocalization with the late endosome and the lysosome was found at 6 and 12 h of the treatment. QDs showed the highest colocalization with late endosomes and lysosomes 24 h after the treatment [31]. Their results suggested that QDs were translocated to the late endosome and the lysosome from 6 to 24 h after the exposure. Although we did not analyze the colocalization of QD with endocytic organelles before 24 h of QD treatment, it is predicted that QDs must be destined toward the early endosome much earlier than 24 h. Overall, it is safe to conclude that most cells can take up QDs within a few hours of their treatment. 

Although culture time is one of the big factors that could affect the viability of QD-treated cells, the amount of QDs internalized by a single cell should also be considered. Several studies reported that a longer culture time than 6 h could lead to unreliable results in QDs tracking analysis because the cell division may result in a dilution effect with the fixed amount of originally treated QDs, only if the provided QD concentration does not affect cell viability [44,45,46]. Likewise, it has been reported that the cell number throughout cell culture with QDs should also be considered. When the cell number increases, the surface of the cell exposed to QDs decreases, thus leading to a lower uptake rate of QDs per cell [42]. As mentioned above, HeLa cells have a much faster doubling time than ML-1 cells. Therefore, the cell number of HeLa should be much higher than that of ML-1 cells under 24 h incubation in case QDs do not affect cell viability, which could cause less QD association with HeLa cells, thus possibly triggering less uptake of QDs per Hela cell. However, although the ICP-OES experiment showed that even though HeLa cells have a fast doubling time, the cell number of QD-treated HeLa was still decreased and more QDs were internalized by QD-treated HeLa than by QD-treated ML-1 cells. This result indicates more QDs internalized by each single HeLa cell.

Although many efforts have been made to elucidate the intracellular trafficking kinetics and localization of QDs and their related internal and external factors, little attention has been paid to the discharge of internalized QDs. In some studies, even though the exocytosis of QDs has been mentioned, the potential impact and mechanism of QD exocytosis were not fully explored [44,47,48,49,50,51,52]. In addition, some previous studies just focused on each part of the intracellular endocytic, trafficking, or exocytosis process of QDs [11,45,48,50,53], but there is no comprehensive investigation of the entire process from entering to exiting the cell by QDs across the cell membrane. In fact, the exocytosis rate of QDs is directly associated with the retention time of QDs, which can significantly affect the cytotoxicity of QDs [54,55,56]. Therefore, it is necessary to comprehensively investigate the complete traffic process of QDs for aiding in further understanding of the interaction of QDs with cells, including treatment doses, time, mechanisms of QDs’ intracellular endocytosis and exocytosis, and intracellular trafficking of QDs.

## 5. Conclusions

The findings of the present study provide the scientific basis for future studies to determine if the rate of intracellular QD kinetics is linked to the severity of cell toxicity in different cell types. The observed intracellular transportation of green CdSe/ZnS QDs in ML-1 and HeLa cells revealed the importance of understanding how the physicochemical properties and surrounding environment of QDs affect the differences in transporting kinetics of QDs and how the interactions between QDs and cells affect changes in toxicities from a kinetics point of view. HeLa cells were found to be more sensitive to CdSe/ZnS and InP/ZnS QDs compared to ML-1 cells, as shown by the significant reduction of HeLa cell viability and increase in apoptosis. Consistently, the trafficking rates of green carboxylated CdSe/ZnS QDs towards the lysosome in HeLa cells were faster than in ML-1 cells, especially when HeLa cells are in the media without serum. Therefore, we conclude that the higher toxicity of CdSe/ZnS QD in HeLa cells is positively correlated with its faster trafficking rate in HeLa cells.

## Figures and Tables

**Figure 1 nanomaterials-12-01517-f001:**
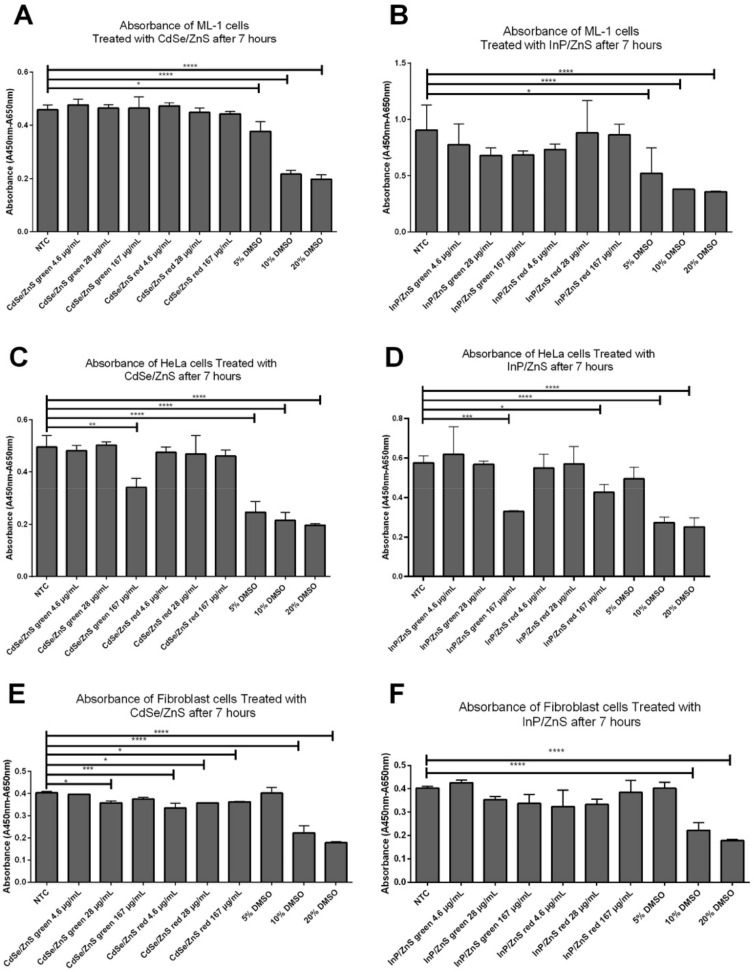
Effects of CdSe/ZnS and InP/ZnS QDs on cell viability in ML-1 and HeLa cell lines measured by XTT reagents. ML-1 thyroid cancer cells or HeLa cells were cultured with varying doses of green and red CdSe/ZnS or InP/ZnS QDs for 24 h, then cell proliferation/viability levels were measured at each concentration using a spectrophotometer at 7 h after XTT treatment. (**A**) ML-1 thyroid cancer cells showed no significant reduction in cell viability at all concentrations of green or red CdSe/ZnS QDs. NTC, non-treated control. DMSO, positive control. (**B**) There was no significant decrease in ML-1 thyroid cancer cells at all concentrations of green or red InP/ZnS QDs. (**C**) Reduced viability was observed in HeLa cells at 167 µg/mL of green CdSe/ZnS QD treatment. (**D**) HeLa cells show a reduction in cell viability at 167 µg/mL of green InP/ZnS QD treatment and 167 µg/mL of red InP/ZnS QD treatment. (**E**) Cell viability test with green and red CdSe/ZnS in mouse-derived non-cancerous fibroblast cells. (**F**) Cell viability test with green and red InP/ZnS in mouse-derived non-cancerous fibroblast cells. Statistically significant results are indicated based on *p*-values: * = *p* < 0.05, ** = *p* < 0.01, *** = *p* < 0.001, **** = *p* < 0.0001.

**Figure 2 nanomaterials-12-01517-f002:**
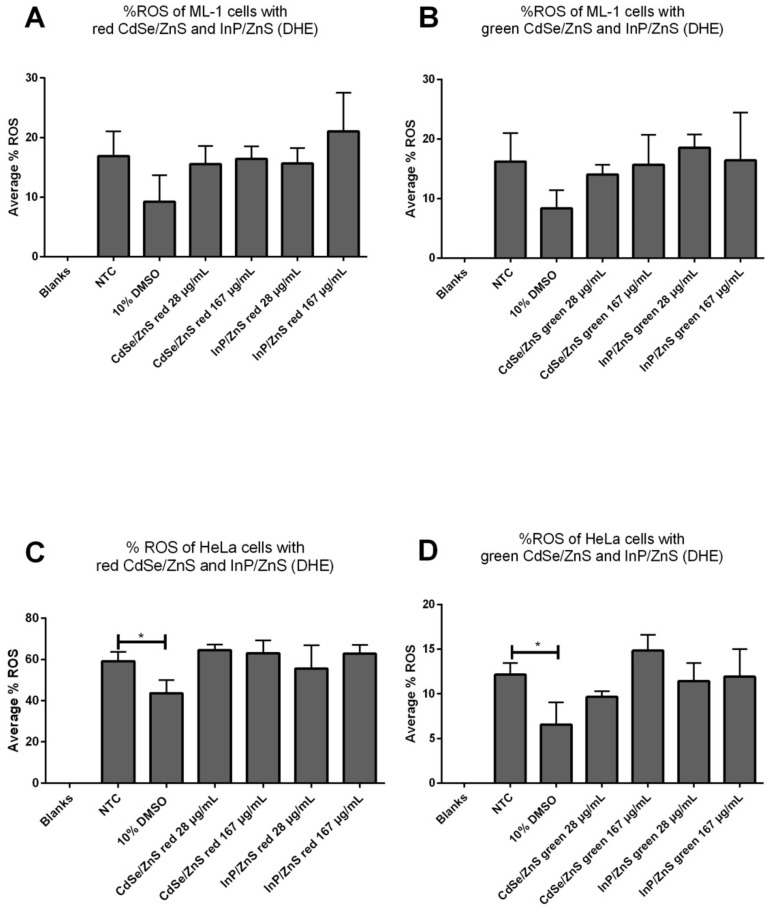
ROS measurements with DHE at varied CdSe/ZnS and InP/ZnS QD concentrations. (**A**) ML-1 cells were treated with 28 µg/mL and 167 µg/mL of red CdSe/ZnS and InP/ZnS QDs for 24 h. (**B**) ML-1 cells were treated with 28 µg/mL and 167 µg/mL of green CdSe/ZnS and InP/ZnS QDs for 24 h. (**C**) HeLa cells were treated with 28 µg/mL and 167 µg/mL of red CdSe/ZnS and InP/ZnS QDs for 24 h. (**D**) HeLa cells were treated with 28 µg/mL and 167 µg/mL of green CdSe/ZnS and InP/ZnS QDs for 24 h. Statistically significant results are indicated based on *p*-values: * = *p* < 0.05.

**Figure 3 nanomaterials-12-01517-f003:**
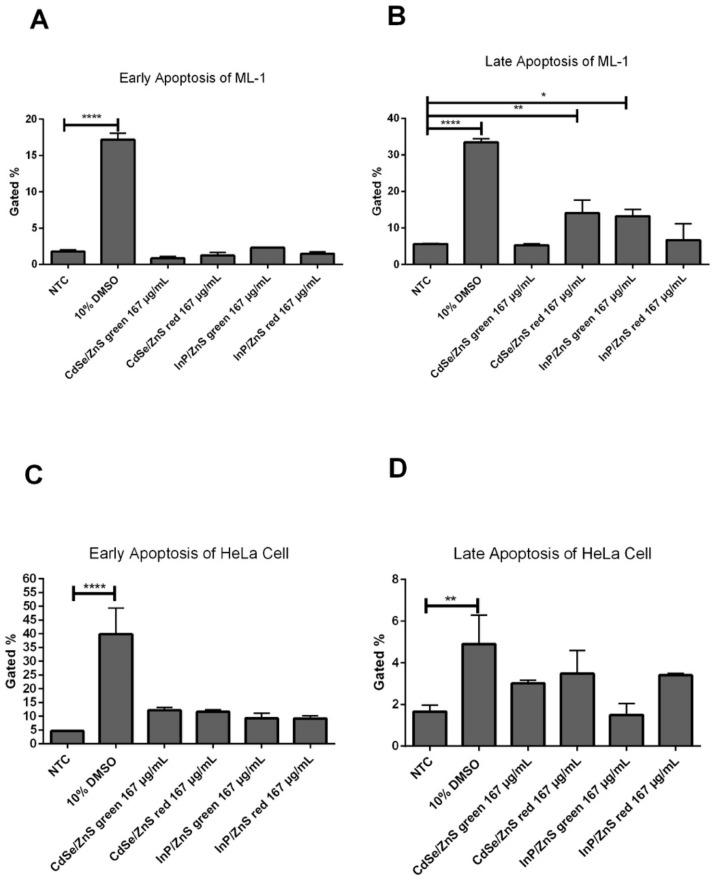
Levels of early and late apoptosis after 24 h treatment of green and red CdSe/ZnS and InP/ZnS QDs. (**A**) Early apoptosis levels of ML-1 cells were recorded by a flow cytometer in samples treated with 167 µg/mL of red CdSe/ZnS and InP/ZnS QDs for 24 h. (**B**) Late apoptosis levels of ML-1 cells were recorded by a flow cytometer in samples treated with 167 µg/mL of red CdSe/ZnS and InP/ZnS QDs for 24 h. (**C**) Early apoptosis levels of HeLa cells were recorded by a flow cytometer in samples treated with 167 µg/mL of red CdSe/ZnS and InP/ZnS QDs for 24 h. (**D**) Late apoptosis levels of HeLa cells were recorded by a flow cytometer in samples treated with 167 µg/mL of red CdSe/ZnS and InP/ZnS QDs for 24 h. Statistically significant results are indicated based on *p*-values: * = *p* < 0.05, ** = *p* < 0.01, **** = *p* < 0.0001.

**Figure 4 nanomaterials-12-01517-f004:**
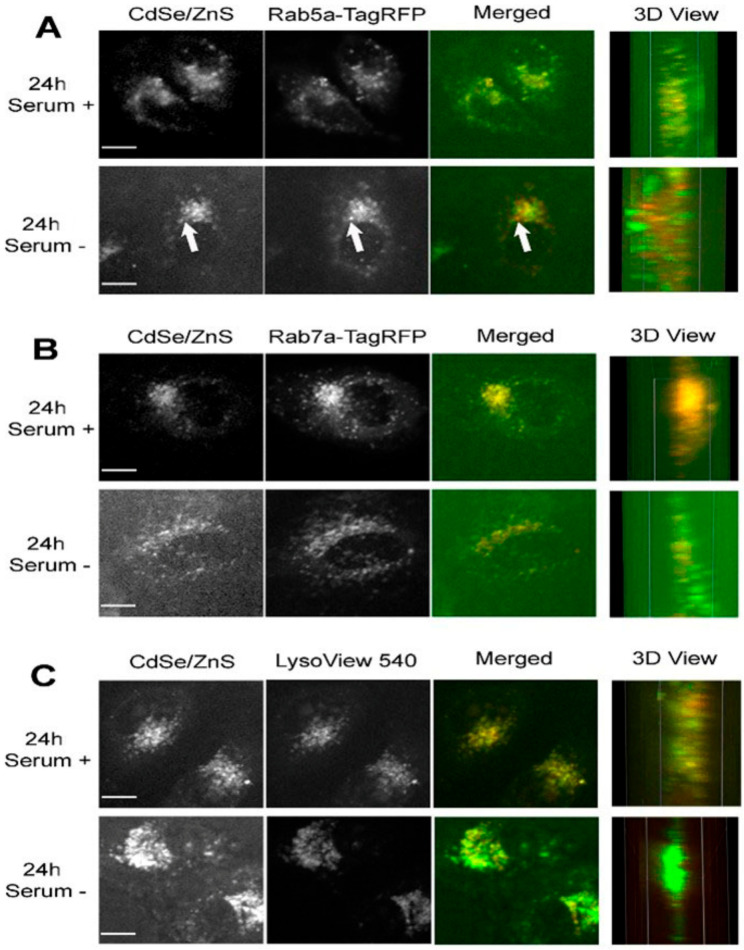
Representative confocal microscope images depicting subcellular localizations of green CdSe/ZnS-COOH QDs after 24 h treatment with QDs in HeLa cell culture. (**A**, **top**) Colocalization of the early endosome reference marker Rab5a-TagRFP with the QDs in the presence of serum. (**A**, **bottom**) Partial colocalization of QDs with Rab5a-TagRFP in cells grown in the medium lacking serum. The arrow indicates a spatial area of no colocalization between QDs and Rab5a-TagRFP. (**B**, **top**) Colocalization of the late endosome reference marker Rab7a-TagRFP with the QDs in the presence of serum. (**B**, **bottom**) Colocalization of QDs with Rab7a-TagRFP in cells grown in the medium lacking serum. (**C**, **top**) Colocalization of the lysosome reference marker LysoView 540 with the QDs in the presence of serum. (**C**, **bottom**) Colocalization of QDs with LysoView 540 in cells grown in the medium lacking serum. Scale bars correspond to 10 µm.

**Figure 5 nanomaterials-12-01517-f005:**
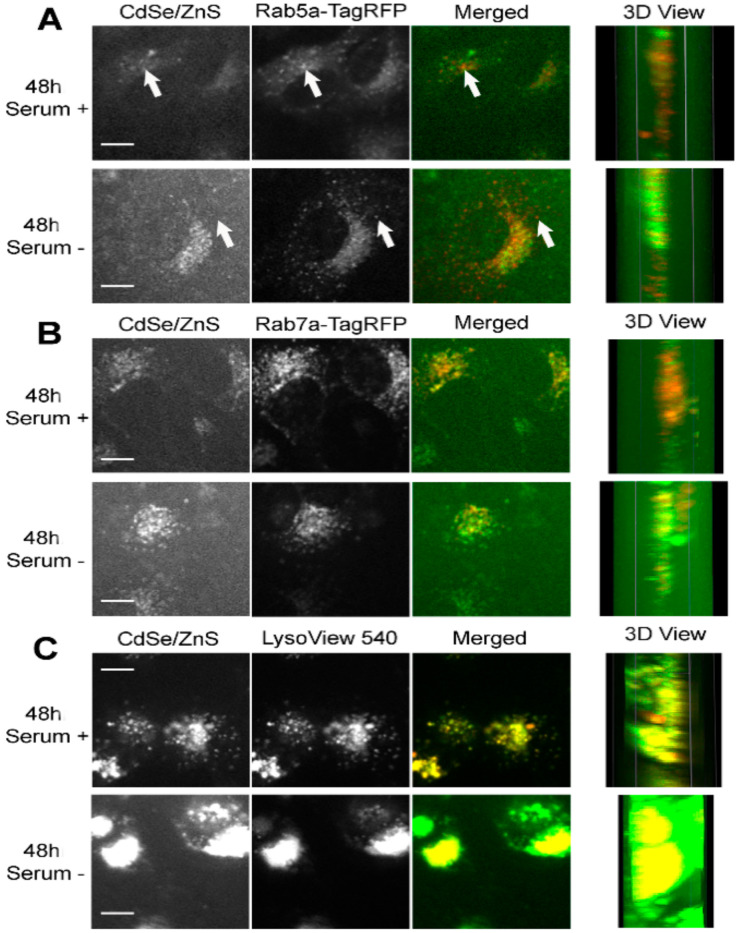
Representative confocal microscope images depicting subcellular localizations of green CdSe/ZnS-COOH QDs after 48 h treatment with QDs in HeLa cell culture. (**A**, **top**) Partial colocalization of the early endosome reference marker Rab5a-TagRFP with the QDs in the presence of serum. (**A**, **bottom**) Partial colocalization of QDs with Rab5a-TagRFP in cells grown in the medium lacking serum. Arrows indicate spatial areas of no colocalization between QDs and Rab5a-TagRFP. (**B**, **top**) Colocalization of the late endosome reference marker Rab7a-TagRFP with the QDs in the presence of serum. (**B**, **bottom**) Colocalization of QDs with Rab7a-TagRFP in cells grown in the medium lacking serum. (**C**, **top**) Colocalization of the lysosome reference marker LysoView 540 with the QDs in the presence of serum. (**C**, **bottom**) Colocalization of QDs with LysoView 540 in cells grown in the medium lacking serum. Scale bars correspond to 10 µm.

**Figure 6 nanomaterials-12-01517-f006:**
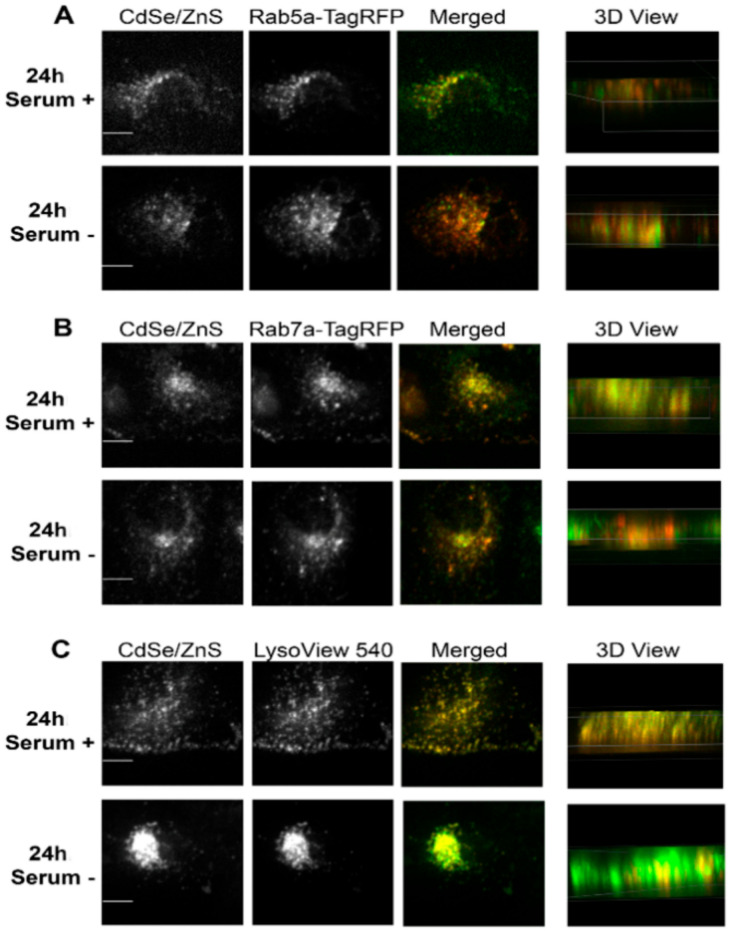
Representative confocal microscope images depicting subcellular localizations of green CdSe/ZnS-COOH QDs after 24 h treatment with QDs in ML-1 cell culture. (**A**, **top**) Colocalization of the early endosome reference marker Rab5a-TagRFP with the QDs in the presence of serum. (**A**, **bottom**) Colocalization of QDs with Rab5a-TagRFP in cells grown in the medium lacking serum. (**B**, **top**) Colocalization of the late endosome reference marker Rab7a-TagRFP with the QDs in the presence of serum. (**B**, **bottom**) Colocalization of QDs with Rab7a-TagRFP in cells grown in the medium lacking serum. (**C**, **top**) Colocalization of the lysosome reference marker LysoView 540 with the QDs in the presence of serum. (**C**, **bottom**) Colocalization of QDs with LysoView 540 in cells grown in the medium lacking serum. Scale bars correspond to 10 µm.

**Figure 7 nanomaterials-12-01517-f007:**
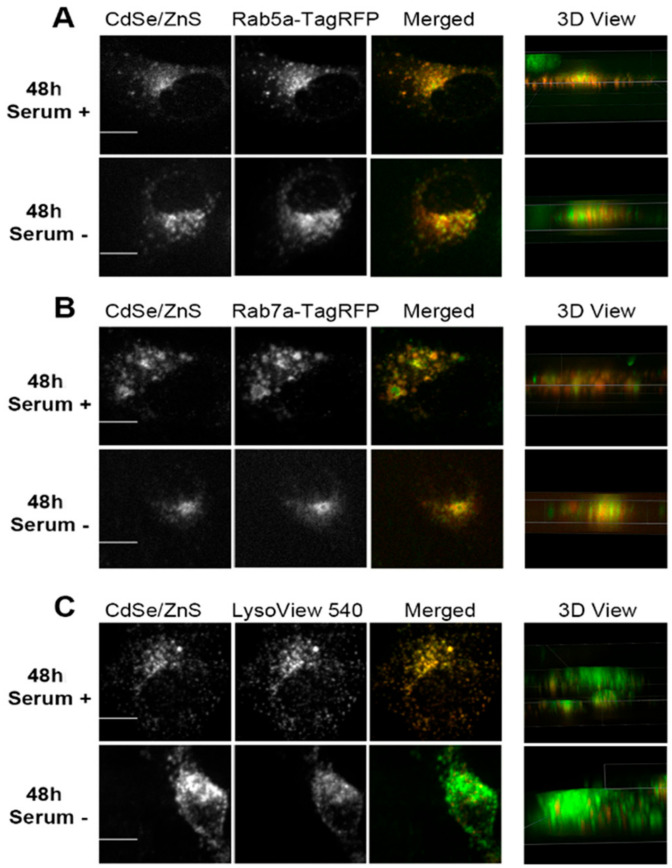
Representative confocal microscope images depicting subcellular localizations of green CdSe/ZnS-COOH QDs after 48 h treatment with QDs in ML-1 cell culture. (**A**, **top**) Colocalization of the early endosome reference marker Rab5a-TagRFP with the QDs in the presence of serum. (**A**, **bottom**) Colocalization of QDs with Rab5a-TagRFP in cells grown in the medium lacking serum. (**B**, **top**) Colocalization of the late endosome reference marker Rab7a-TagRFP with the QDs in the presence of serum. (**B**, **bottom**) Colocalization of QDs with Rab7a-TagRFP in cells grown in the medium lacking serum. (**C**, **top**) Colocalization of the lysosome reference marker LysoView 540 with the QDs in the presence of serum. (**C**, **bottom**) Colocalization of QDs with LysoView 540 in cells grown in the medium lacking serum. Scale bars correspond to 10 µm.

**Figure 8 nanomaterials-12-01517-f008:**
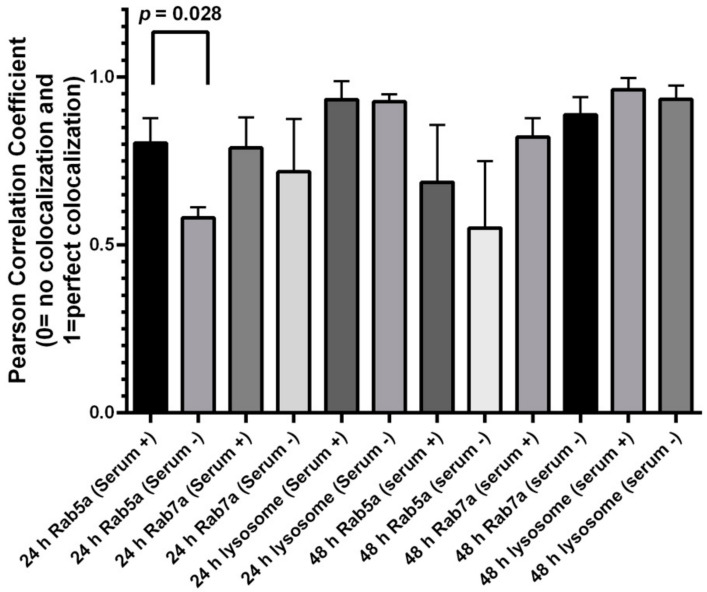
Colocalization assay depicting the level of colocalization between CdSe/ZnS QD and fluorescence organelle markers in HeLa cell culture. Graph bars represent the results of colocalization analysis using the JACoP plugin from ImageJ using Pearson’s correlation coefficient. Mean Pearson coefficient and standard deviation values for levels of colocalization between QDs and fluorescent organelle markers are indicated in the graph. The mean Pearson coefficient value for the level of colocalization between QDs and the early endosome in the culture with serum was statistically significantly higher than that for cells grown in the culture lacking serum (*p* = 0.028, via Prism GraphPad Dunnett test).

**Figure 9 nanomaterials-12-01517-f009:**
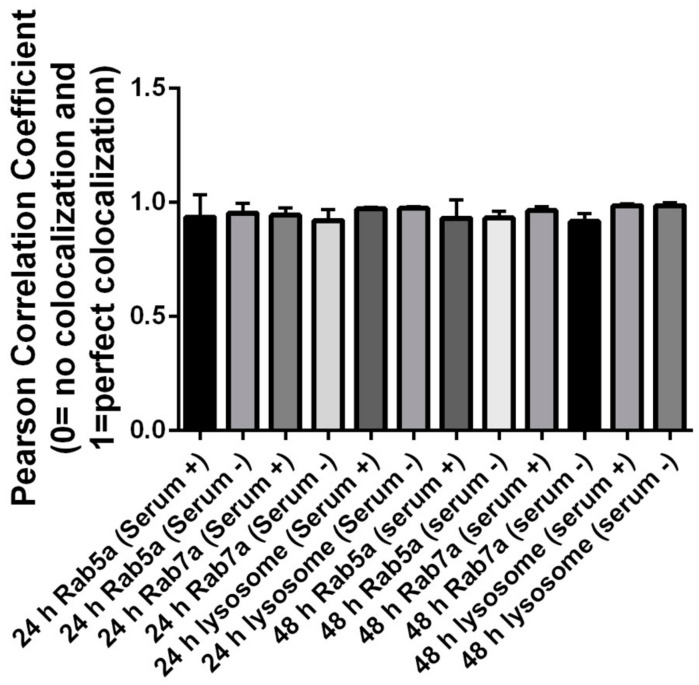
Colocalization assay depicting the level of colocalization between CdSe/ZnS QD and fluorescence organelle markers in ML-1 cell culture. Graph bars represent the results of colocalization analysis using the JACoP plugin from ImageJ using Pearson’s correlation coefficient. Mean Pearson coefficient and standard deviation values for levels of colocalization between QDs and fluorescent organelle markers are indicated in the graph.

**Figure 10 nanomaterials-12-01517-f010:**
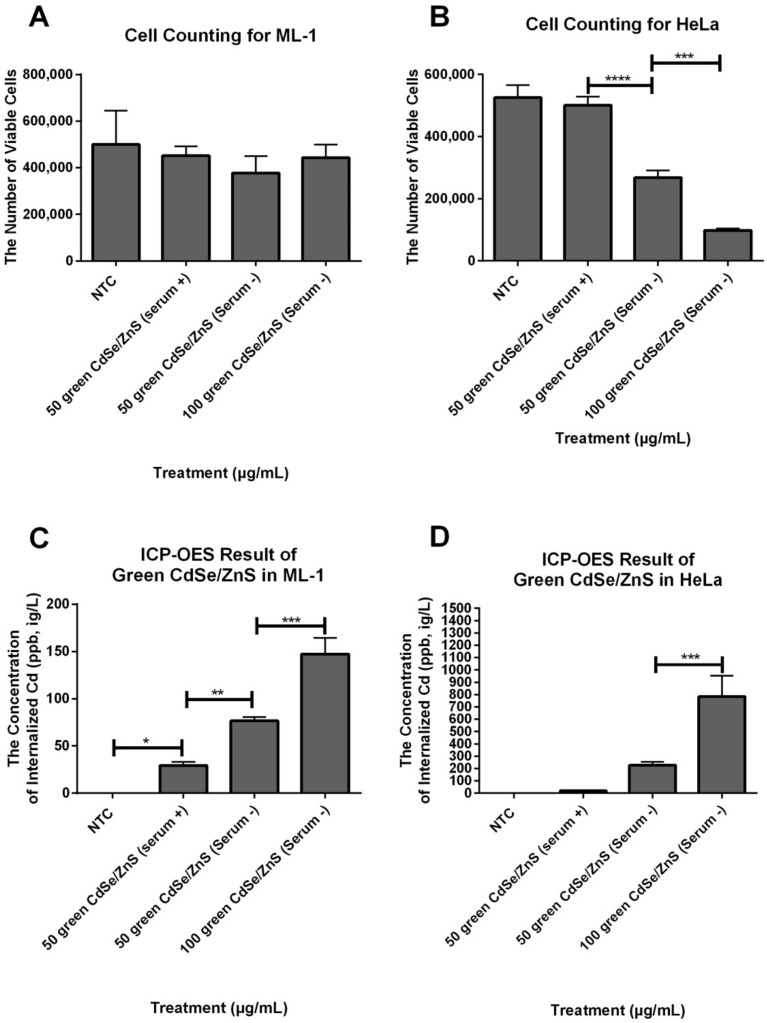
The concentration of internalized QDs by ML-1 and HeLa cells treated with 50 and 100 µg/mL green CdSe/ZnS in serum+ and −mediums for 48 h. (**A**) The number of viable ML-1 cells that were treated with 50 and 100 µg/mL green CdSe/ZnS QDs in serum+ and −mediums after 48 h. (**B**) The number of viable HeLa cells that were treated with 50 and 100 µg/mL green CdSe/ZnS QDs in serum+ and −mediums after 48 h. (**C**) The concentration of internalized QDs by ML-1 that were treated with 50 and 100 µg/mL green CdSe/ZnS QDs in serum+ and −mediums after 48 h. (**D**) The concentration of internalized Cd by HeLa that were treated with 50 and 100 µg/mL green CdSe/ZnS QDs in serum+ and − media for 48 h. Statistically significant results are indicated based on *p*-values: * = *p* < 0.0133, ** = *p* < 0.01, *** = *p* < 0.001, **** = *p* < 0.0001.

**Figure 11 nanomaterials-12-01517-f011:**
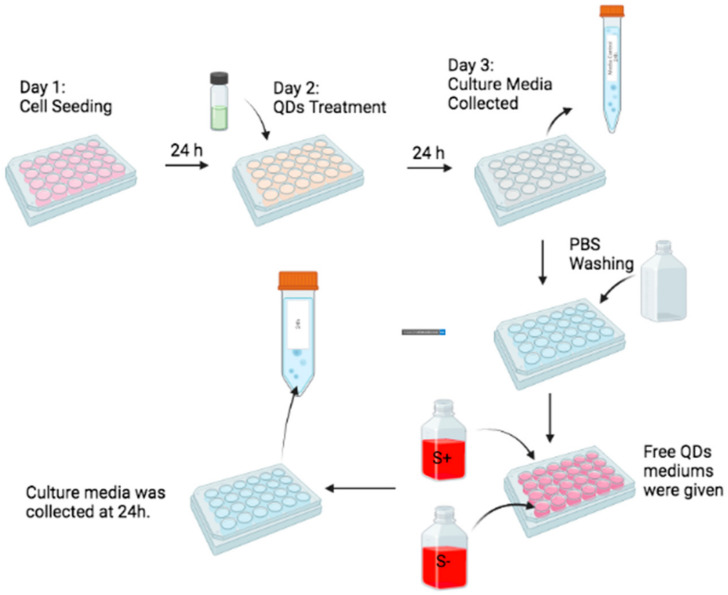
A figure illustrating the ICP-OES method for quantifying the amount of internalized and released QDs under different treatment conditions: 50 µg/mL of green CdSe/ZnS QDs with or without serum as well as cells with 100 µg/mL cultured in serum− media for 24 h.

**Figure 12 nanomaterials-12-01517-f012:**
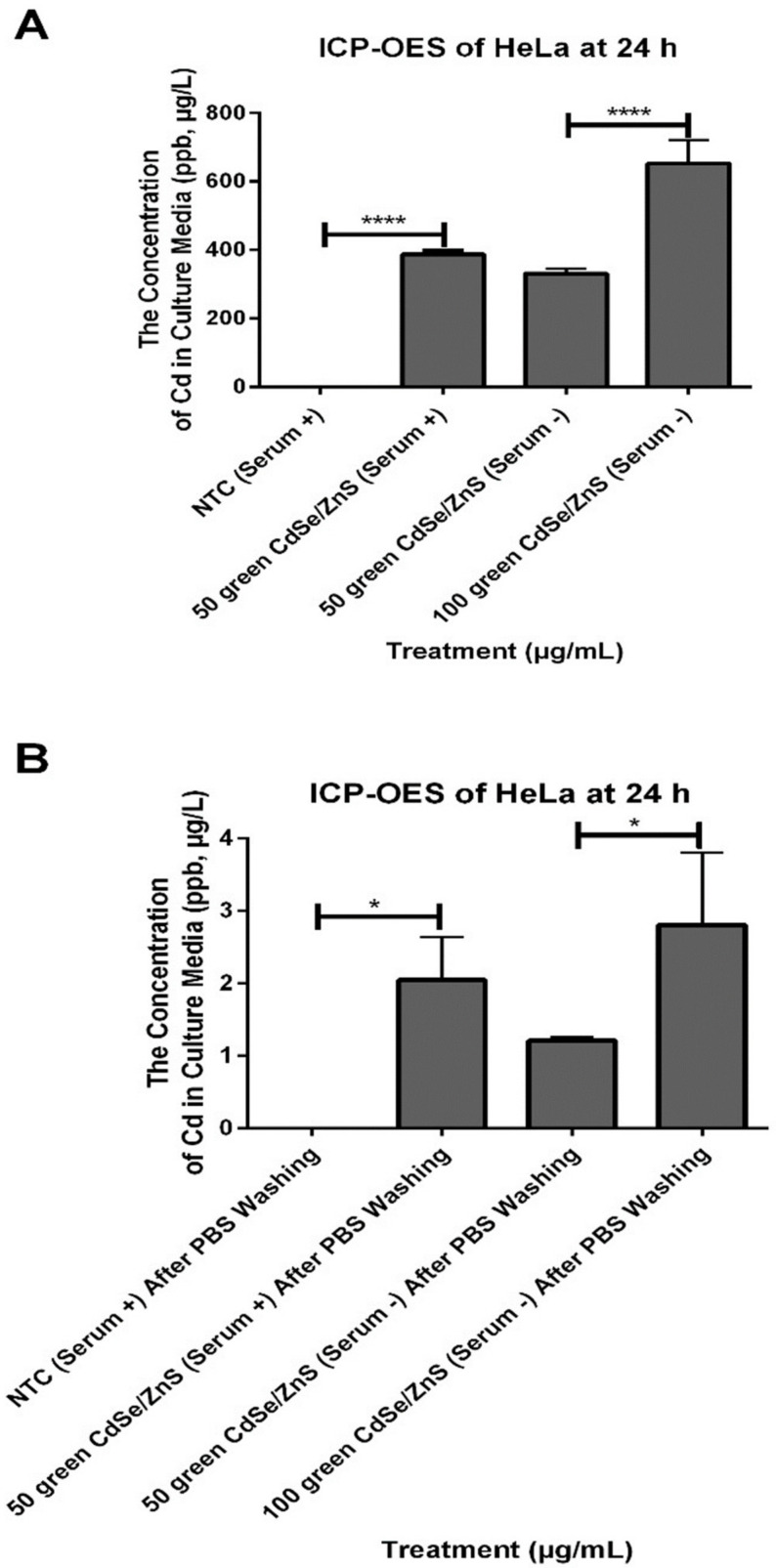
The concentration of secreted QDs by ML-1 and HeLa cells treated with 50 and 100 µg/mL green CdSe/ZnS in serum+ and −mediums for 24 h. (**A**) The concentration QDs in the culture media of HeLa that were treated with 50 and 100 µg/mL green CdSe/ZnS QDs in serum+ and −mediums at 24 h. (**B**) The concentration of secreted QDs by HeLa that were treated with 50 and 100 µg/mL green CdSe/ZnS QDs in serum+ and −mediums at 24 h. Statistically significant results are indicated based on *p*-values: * = *p* < 0.05, **** = *p* < 0.0001.

**Figure 13 nanomaterials-12-01517-f013:**
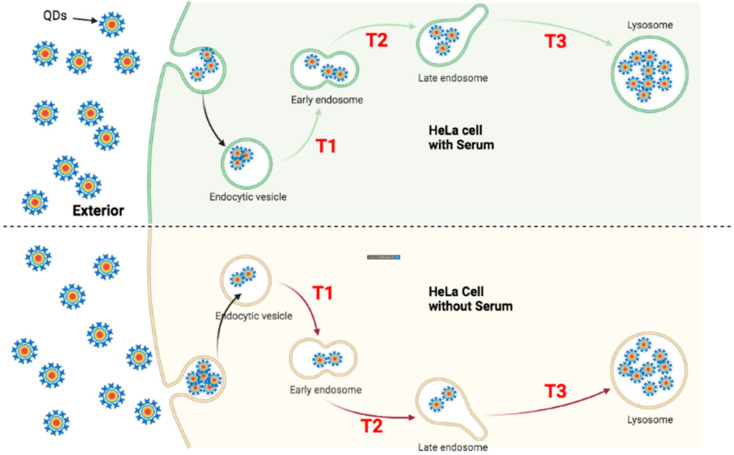
Proposed transportation kinetics of QDs in HeLa cells grown in the media with and without serum for 24 h. The models depict that the intracellular traffic kinetics of QDs in HeLa cells grown with serum (**Upper**) is noticeably distinctive from that of HeLa cells grown without serum (**Bottom**) after 24 h of QD treatment. (T1) indicates the transition of QDs from the endocytic vesicle towards the early endosome; (T2) indicates the transition of QDs from the early endosome towards the late endosome; (T3) indicates the transition of QDs from the late endosome to the lysosome. Each green arrow represents a slower transit rate of QDs in comparison with its counterpart rate indicated in red. Each red arrow represents a faster transit rate of QDs in comparison with its counterpart rate indicated in green.

**Figure 14 nanomaterials-12-01517-f014:**
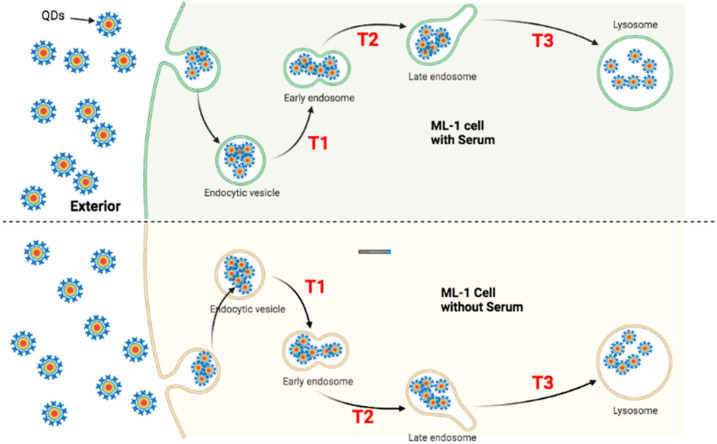
Proposed transportation kinetics of QDs in ML-1 cells grown in the media with and without serum for 24 h. The models depict that the intracellular traffic kinetics of QDs in ML-1 cells grown with serum (**Upper**) is noticeably distinctive from that of ML-1 cells grown without serum (**Bottom**) after 24 h of QD treatment. (T1) indicates the transition of QDs from the endocytic vesicle towards the early endosome; (T2) indicates the transition of QDs from the early endosome towards the late endosome; (T3) indicates the transition of QDs from the late endosome to the lysosome.

**Figure 15 nanomaterials-12-01517-f015:**
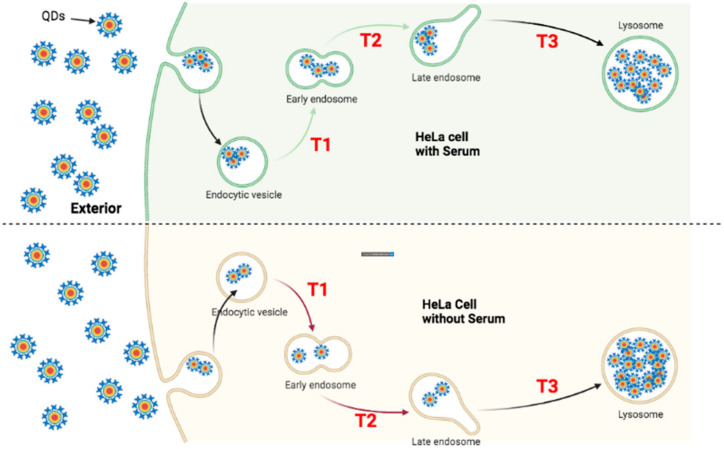
Proposed transportation kinetics of QDs in HeLa cells grown in the media with and without serum for 48 h. A figure illustrating intracellular trafficking kinetics of QDs varies in HeLa cells grown with serum (**Upper**) and HeLa cells grown without serum (**Bottom**) 48 h after the treatment of QDs. (T1) indicates the transition of QDs from the endocytic vesicle towards the early endosome; (T2) indicates the transition of QDs from the early endosome towards the late endosome; (T3) indicates the transition of QDs from the late endosome to the lysosome. Each green arrow represents a slower transit rate of QDs in comparison with its counterpart rate indicated in red. Each red arrow represents a faster transit rate of QDs in comparison with its counterpart rate indicated in green.

**Figure 16 nanomaterials-12-01517-f016:**
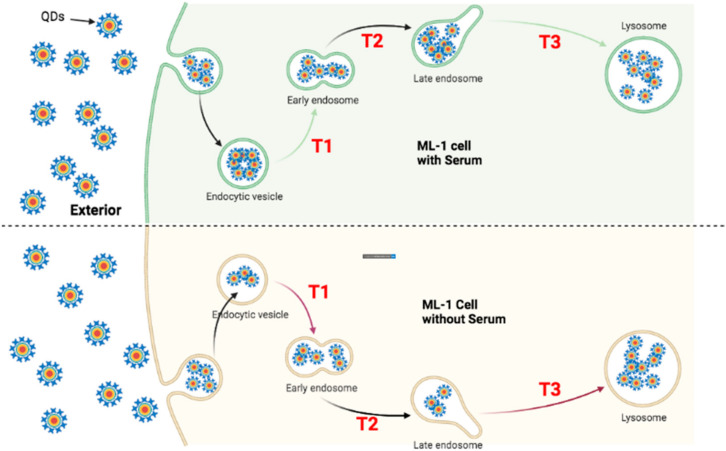
Proposed transportation kinetics of QDs in ML-1 cells grown in media with and without serum for 48 h. A figure illustrating intracellular trafficking kinetics of QDs varies in ML-1 cells grown with serum (**Upper**) and ML-1 cells grown without serum (**Bottom**) cells 48 h after the treatment of QDs. (T1) indicates the transition of QDs from the endocytic vesicle towards the early endosome; (T2) indicates the transition of QDs from the early endosome towards the late endosome; (T3) indicates the transition of QDs from the late endosome to the lysosome. Each green arrow represents a slower transit rate of QDs in comparison with its counterpart rate indicated in red. Each red arrow represents a faster transit rate of QDs in comparison with its counterpart rate indicated in green.

**Table 1 nanomaterials-12-01517-t001:** IC50 values and ranking of toxicity of green and red CdSe/ZnS and InP/ZnS QDs in Hela cells.

	Size of QDs (from Smallest to Largest)	IC50	Ranking of Toxicity
Green InP/ZnS	3.7–5.2 nm	97 µg/mL	1
Green CdSe/ZnS	6.1–9.5 nm	143 µg/mL	3
Red CdSe/ZnS	5–10 nm	167 µg/mL	4
Red InP/ZnS	10–20 nm	139 µg/mL	2

## Data Availability

All data are included in the manuscript and Appendix A.

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
