# Peer review of "Intracellular Trafficking and Distribution of Cd and InP Quantum Dots in HeLa and ML-1 Thyroid Cancer Cells"

_nanomaterials, 2022, doi:10.3390/nano12091517_

Round 1

Reviewer 1 Report

In this manuscript, the author choose CdSe/ZnS and InP/ZnS QDs to present study aims to elucidate the toxicity and intracellular transport kinetics. And, the ICP-OES test showed the uptake of CdSe/ZnS QDs in HeLa cells was significantly higher than in ML-1 cells.I think this research has a lot of implications for quantum dot bioimaging. Considering the present manuscript, it can be recommended for publication in the nanomaterials, after minor revision.

  1. Why did you choose these cell models?
  2. Are these quantum dots synthetic or purchased?
  3. The advantage ofCdSe/ZnS and InP/ZnS QDs are that they are resistant to photobleaching,please add relevant references on their application in biological imaging(super resolution imaging)
  4. Please add a scale value in the cell image .
  5. Please add relevant optical characterization dataof  CdSe/ZnS and InP/ZnS QD.

Author Response

We have completed the revision requested by the reviewer and attached a rebuttal letter for that and a revised manuscript as well as a supplementary document.

If you have any questions, please let me know.

Thank you for your consideration of this manuscript.

Sincerely,

Kyoungtae Kim

Professor of Biology

Missouri State University

[email protected]

Reviewer 2 Report

The present study lacks novelty. Please note the effects of Cd QDs on a number of cell lines including HeLa cells were already reported. The rationale of the study is also confusing: “The present study aims to elucidate the toxicity and intracellular transport kinetics of CdSe/ZnS and InP/ZnS QDs in late-stage ML-1 thyroid cancer using well-tested HeLa as a control”. Why cervical cancer cells are control cells for thyroid cancer cells? The authors showed and commented that “ML-1 cells exhibit no viability defect in response to QDs, whereas HeLa cell viability decreases. These results suggest that HeLa cells are more sensitive to QDs compared to ML-1 cells”. The authors also correlated QD-mediated cytotoxicity with QD uptake and concluded that “the higher toxicity of CdSe/ZnS QD in HeLa cells is positively correlated with its faster trafficking rate in HeLa cells”.

The present study is just a technical note and this is not a comprehensive original research report. The study is purely descriptive. No molecular mechanism of cytotoxic action is provided. Scientific significance is limited.

Author Response

(The authors gave the same response as above.)

Reviewer 3 Report

This manuscript was studied the cytotoxicity of CdSe/ZnS and InP/ZnS QDs in ML-1 thyroid cancer cells. The CdSe/ZnS and InP/ZnS QDs intracellular endocytic trafficking kinetics were further explored in ML-1 cells and HeLa cell. The mothed has provided new insights for effective cancer therapy. Although lots of data have been provided to demonstrate the interaction of engineered nanoparticles, several issues listed as follows need to be seriously considered and minor revision before its publication:

  1. As described by authors, CdSe/ZnS and InP/ZnS QDs play an important role in this article. Therefore, it should be characterized by proper methods to demonstrate the morphologies and sizes, such as TEM, SEM.
  2. Abstract: Page 1, Line no 6: XTT shall be expanded as majority of the readers won't understand what is XTT. expand the term for better clarity.
  3. Why was 10%DMSO used instead of 20%DMSO used as a control in Figure 2?
  4. The labels of DMEM+ serum and DMEM- serum in the before and after pictures are inconsistent.(Fig8 and Fig10)

Author Response

(The authors gave the same response as above.)

Round 2

Reviewer 2 Report

The manuscript was improved.